# Cognitive function and treatment response trajectories in first-episode schizophrenia: evidence from a prospective cohort study

Edward Millgate ![ORCID],[1] Kira Griffiths,[1] Alice Egerton,[1,2] Eugenia Kravariti,[1,2] Cecilia Casetta,[1,3] Bill Deakin,[4,5] Richard Drake ![ORCID],[4,5] Oliver D Howes,[1,2] Laura Kassoumeri,[1] Sobia Khan,[4] Steve Lankshear,[5] Jane Lees,[4,5] Shon Lewis,[4,5] Elena Mikulskaya,[4] Ebenezer Oloyede,[1,6] Rebecca Owens,[4] Rebecca Pollard ![ORCID],[1] Nathalie Rich,[7] Sophie Smart,[8] Aviv Segev,[1,9] Kyra Verena Sendt,[1] James MacCabe,[1,3] The STRATA Consortium

For numbered affiliations see end of article.

**Correspondence to**
Dr James MacCabe;
james.maccabe@kcl.ac.uk

## ABSTRACT

**Objectives** This prospective cohort study tested for associations between baseline cognitive performance in individuals early within their first episode and antipsychotic treatment of psychosis. We hypothesised that poorer cognitive functioning at the initial assessment would be associated with poorer antipsychotic response following the subsequent 6 weeks.

**Design** Prospective cohort .

**Setting** National Health Service users with a first-episode schizophrenia diagnosis, recently starting antipsychotic medication, recruited from two UK sites (King's College London, UK and University of Manchester, UK). Participants attended three study visits following screening.

**Participants** Eighty-nine participants were recruited, with 46 included in the main analysis. Participants required to be within the first 2 years of illness onset, had received minimal antipsychotic treatment, have the capacity to provide consent, and be able to read and write in English. Participants were excluded if they met remission criteria or showed mild to no symptoms.

**Primary and secondary outcome measures** Antipsychotic response was determined at 6 weeks using the Positive and Negative Syndrome Scale (PANSS), with cognitive performance assessed at each visit using the Brief Assessment of Cognition in Schizophrenia (BACS). The groups identified (responders and non-responders) from trajectory analyses, as well as from >20% PANSS criteria, were compared on baseline BACS performance.

**Results** Trajectory analyses identified 84.78% of the sample as treatment responsive, and the remaining 15.22% as treatment non-responsive. Unadjusted and adjusted logistic regressions observed no significant relationship between baseline BACS on subscale and total performance (BACS t-score: OR=0.98, p=0.620, Cohen's d=0.218) and antipsychotic response at 6 weeks.

**Conclusions** This investigation identified two clear trajectories of treatment response in the first 6 weeks of antipsychotic treatment. Responder and non-responder groups did not significantly differ on performance on the

### STRENGTHS AND LIMITATIONS OF THIS STUDY

⇒ The study examined cognitive performance in a multicentre sample of first-episode psychosis.
⇒ Cognitive performance was assessed at each study period with the Brief Assessment of Cognition in Schizophrenia, a reliable and well-validated brief test battery which is specifically designed for schizophrenia.
⇒ The study used symptom ratings on the Positive and Negative Syndrome Scale to determine response to treatment, a gold-standard proxy within the current research field.

BACS, suggesting that larger samples may be required or that an association between cognitive performance and antipsychotic response is not observable in the first 2 years of illness onset.

**Trial registration number** REC: 17/NI/0209.

## INTRODUCTION

Prompt intervention with pharmacological therapy for individuals with schizophrenia has been extensively recommended in the literature[1 2] and is reported to be associated with better functional outcomes.[3–5] As observed by Carbon and Correll,[5] a lack of early response and improvement to antipsychotic medication is a strong predictor of later non-response. A recent diagnostic test review has even argued that non/minimal response to antipsychotic medication in the first 2 weeks of treatment may be a sufficient indication to switch antipsychotic.[6] Early and accurate detection of treatment non-responders at first episode is also more likely to result in timely treatment with clozapine, which may be associated with better outcomes.[7] Indeed, Yoshimura *et al*[8]

found that response to clozapine was ~80% in treatment-resistant patients who were commenced on clozapine early in their illness course, with this depreciating to ~30% when clozapine initiation was delayed by more than 2.8 years.[7 8]

Individuals who do not respond to antipsychotic medication are reported to have higher rates of smoking (56%), substance and alcohol abuse (51%) and suicidal ideation (44%), with annual treatment costs being 3–11 times larger than those who respond to antipsychotic medication.[9] In 2007, it was estimated that schizophrenia accounted for 30% of the total expenditure for adult mental health and social care services,[10] with additional economic and societal costs due to unemployment or absence from work. These total service costs, which were estimated at £2.2 billion in 2007, have the potential to reach £3.7 billion by 2026.[11] However, it has been suggested that early intervention programmes could aid in reducing these costs substantially if adequately introduced in first-episode psychosis,[12] as earlier-onset schizophrenia is associated with greater expected costs.[11]

Early cognitive deficits may be predictive of subsequent antipsychotic response in the first episode of illness and could aid in delivering fast, early intervention. Cognitive dysfunction in schizophrenia is observable prior to illness onset[13 14] and is strongly associated with poorer functional outcomes.[15–17] A recent meta-analysis comparing cognitive performance in known cases of antipsychotic treatment resistance and response[18] observed worse performance in treatment-resistant samples across cognitive domains, with the strongest effect in measures of verbal memory and learning and language functions. However, it is possible that illness chronicity and exposure to long-term antipsychotic treatment may have influenced these findings.

Based on the current existing literature, it is plausible to argue that there may be quantifiable cognitive differences between individuals who respond to antipsychotic medication and those who do not in the early stages of the illness; seeing as deficits in cognition are observable prior to illness onset[14 15] and poor early non-response to medication being predictive of future non-response.[5] Therefore, if differences are observed between groups of differing response to medication (ie, responders and non-responders), early in their illness and treatment, this will broaden our understanding of the relationship between cognition, schizophrenia and antipsychotic response, as well as aid clinical utility by using brief cognitive measures as a screening for potential non-response in the first episode of schizophrenia. The American Psychological Association's (APA) Working Group on Screening and Assessment has provided guidelines for determining the appropriateness of a neuropsychological measure for cognitive screening within a clinical setting.[19] The guidelines are as follows: (1) provide identification for those at high risk of impairment, (2) sensitive enough to identify those who need further review, (3) brief and narrow in scope, (4) can be administered at routine visits, (5) can be administered by support staff or clinicians electronically, and (6) can be used to monitor progress and outcomes.[20] In high-income countries, the use of brief assessment batteries such as the Brief Assessment of Cognition in Schizophrenia (BACS) has been found to meet these criteria put forward by the APA Working Group.[21]

Therefore, this prospective cohort study tested for associations between baseline cognitive performance using a brief cognitive battery, assessed at the initiation of antipsychotic treatment, in individuals early within their first episode of psychosis and their subsequent response to antipsychotic treatment. We hypothesised that poorer cognitive functioning at the initial assessment would be associated with poorer response over the subsequent 6 weeks of antipsychotic treatment.

## METHODS
### Design
The study used a prospective cohort design with a sample of patients with first-episode schizophrenia. Participants were assessed over a period of 6 weeks, with two follow-up visits following baseline and screening assessments.

### Setting
The study was part of the 'Schizophrenia: Treatment Resistance and Therapeutic Advances' (STRATA) Consortium, which included two UK sites in this study: King's College London (London, UK) and University of Manchester (Manchester, UK). The aim of the STRATA Consortium is to identify neurobiological, cognitive and genetic biomarkers of antipsychotic treatment resistance and non-response within schizophrenia and other related psychotic disorders.

### Patient and public involvement
In the early development and design of the study, consultations with the National Institute for Health Research (NIHR) Maudsley Biomedical Research Centre (BRC) Service User Advisory Group took place to determine the feasibility of the study and its assessments for service users. The NIHR Maudsley BRC Feasibility and Acceptability Support Team for Researchers service was also used in order to receive feedback on consent forms, information sheets and protocols, as well as advice for recruitment strategies for service users.

### Participants
Eighty-nine participants aged between 18 and 65 years with a Diagnostic Statistical Manual 5th edition diagnosis of schizophrenia, schizoaffective, schizophreniform disorder or psychosis (non-specified) (International Classification of Diseases 10th edition : F20–F29) were recruited across two UK sites (King's College London and University of Manchester). Inclusion required that participants were within the first 2 years of illness onset, defined using the date of first initial contact with services and clinical records. Inclusion also required that participants had

received minimal antipsychotic medication, which was defined as having received antipsychotic treatment for no longer than 4 weeks prior to the baseline visit, after a period of being either antipsychotic naïve or antipsychotic free for at least 14 days. Participants were assessed at baseline within the first 2 weeks of antipsychotic medication initiation. Participants were also required to have the capacity to provide consent and the ability to read and write in English. Participants were excluded if they met modified Andreasen remission criteria,[22] having mild or less scores on all of the following Structured Clinical Interview-Positive and Negative Syndrome Scale (SCI-PANSS)[23] items: delusions (P1), conceptual disorganisation (P2), hallucinatory behaviour (P3), blunted affect (N1), social withdrawal (N4), lack of spontaneity (N6), mannerisms/posturing (G5), unusual thought content (G9) on the day of assessment, as this would suggest that their symptoms were in remission. Participants were also required to show adherence to medication, as evidence by a Kemp Clinician Rating Scale (CRS)[24 25] of equal to or greater than 3 ('accepts only because compulsory, or very reluctant/requires persuasion, or questions the need for medication often').

Participants were assessed within the first 14 days of starting antipsychotic medication at baseline, 2 weeks from baseline assessment (±7 days of date) and 6 weeks from baseline assessment (±7 days of date), with the maximum cut-off for 6-week follow-up being 56 days after baseline assessment (ie, if an individual was assessed at the maximum follow-up periods at 2-week and 6-week visits; 8 weeks total). Participants were reimbursed for their time and expenses for participation in the study. Fourteen participants were withdrawn after providing consent, an additional 20 were withdrawn following baseline and another 9 participants withdrawn following 2-week assessment. Participants were withdrawn if they were unable to attend the study visit, their symptoms were in remission (as per Andreasen remission criteria[22]), or if they no longer wanted to take part in the study and requested to have their data removed (see figure 1). Forty-six participants were eligible for inclusion in the analysis. All participants gave informed consent prior to enrolment.

### Definitions for treatment response status
Treatment response groups were modelled through trajectory analyses using the *traj* command in STATA.[26] This tool estimates group-based trajectories over a specified time interval, clustering individuals who follow similar trajectories through a censored normal model. Akaike information criterion (AIC) and Bayesian information criterion (BIC) values were used to select the trajectory model with the lowest AIC and BIC values. Linear trajectories of up to five classes (one to five trajectories) were assessed for eligibility. Rescaled PANSS scores,[27] calculated by subtracting 30 from total scores prior to producing estimates for percentage change at weeks 2 and 6, were used in the model. The results generated using this trajectory grouping were also compared with a more standard definition of treatment response which uses a >20% reduction in rescaled PANSS total scores from initial to final assessment.[28 29] Here, patients not reaching a 20% reduction in rescaled PANSS total scores at the 6-week visits were categorised as non-responsive. These results are reported in the online supplemental material 1.

## MATERIALS
### Clinical and demographic measures
At baseline, participants completed the Kemp CRS,[24 25] Mini-International Neuropsychiatric Interview[30] (M-Psychotic Disorders; A-Major Depressive Episode; D-Manic/Hypomanic/Bipolar), SCI-PANSS[23] and Clinician Rating Scale for Schizophrenia (CGI-SCH),[31] and provided demographic data. At each subsequent study visit, the CRS, SCI-PANSS and CGI-SCH were repeated.

### Neuropsychological assessment
Participants completed version A of the BACS[32] at each study visit. The BACS was originally developed to assess cognitive functioning in schizophrenia, while also being an easily administrable and brief test battery.[32] The battery includes tasks pertaining to executive functions, working memory, motor/processing speed, verbal memory, verbal fluency and attention cognitive domains. Version A includes the following tasks: (1) list learning task (verbal memory); (2) digit sequencing task (working memory); (3) token motor task (motor speed); (4) category instances task (verbal fluency); (5) symbol coding task (attention and speed of information processing) and (6) Tower of London (executive function). All tasks on the BACS are scored such that higher scores represent better performance. Composite t-score and z-score for the BACS were generated using scores from published normative data.[33]

### Data analysis
All analyses were conducted in STATA V.15/SE.[34] Wilcoxon signed-rank tests were used to compare cognitive performance and symptom severity in the whole sample between visits (ie, baseline assessment to 2 weeks, 2–6 weeks and baseline to 6 weeks) as not all symptom severity and cognitive variables were normally distributed. Baseline cognitive performance on the BACS was compared between trajectory groups using multivariable logistic regressions on the BACS composite and subscale scores. All models were adjusted for age, gender and days from first psychotic symptom to baseline antipsychotic medication (ie, duration of untreated psychosis (DUP)). Results were then compared with groupings based on >20% reduction in rescaled PANSS total scores[27 28] from baseline assessment to 6-week follow-up (online supplemental material table 1).

Finally, growth curve models were executed using the *xtmixed* command[35] to compare cognitive performance over time between trajectory groups to estimate any changes in cognitive performance over the study period

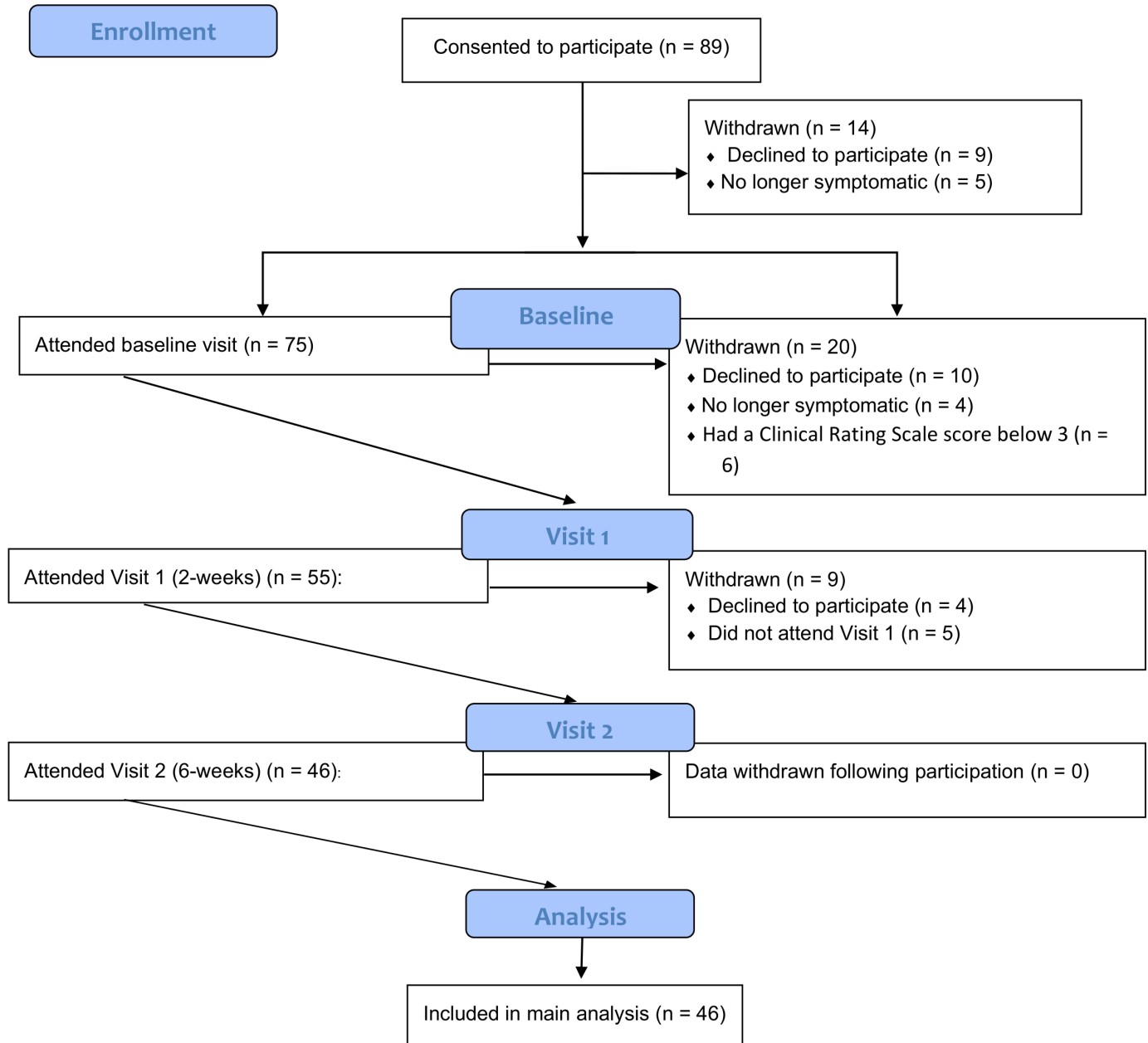

**Figure 1** A CONSORT-based flow chart illustrating the number of participants and exclusions at each stage of the study trial. CONSORT, Consolidated Standards of Reporting Trials.

(online supplemental material table 2). These results were again compared with >20% PANSS reduction criteria for treatment response (online supplemental material table 3).

## RESULTS

Table 1 reports the demographic descriptive of the whole sample included in analysis (N=46) at baseline, with PANSS symptom severity scores and BACS performance for each study visit illustrated in table 2. Data regarding antipsychotic medication were provided by all participants at baseline, all of which were treated with second-generation antipsychotics. At baseline, 45 participants provided PANSS symptom severity ratings, with

41 providing at least one baseline measure of the BACS (table 2). Between baseline and 2-week assessment, the average follow-up was 18.19 days (SD=6.6) and between 2 weeks and 6 weeks, this was 26.69 days (SD=9.6). Between baseline and 6-week visit, the study trial lasted 43.86 days (SD=7.2).

Between study visits, a significant improvement in PANSS positive symptom scores was observed in the whole sample between baseline and 2-week visits, 2-week and 6-week visits, as well as baseline and 6-week assessments (table 2). A significant improvement in PANSS total scores was observed between baseline and 2-week and baseline and 6-week visits. No significant differences in symptom severity were observed between visits for

**Table 1** Descriptive statistics of the whole sample demographics at consent (age) and baseline assessments

| Variable | N | M | SD | Min | Max |
|---|---|---|---|---|---|
| Age (at consent) | 46 | 27.30 | 8.17 | 18 | 50 |
| Gender (male) | 33 (71.74%) | – | – | – | – |
| Gender (female) | 13 (28.26%) | – | – | – | – |
| Age of illness onset (years) | 46 | 26.53 | 8.45 | 18 | 49 |
| Duration from 1st psychotic symptom (days) to baseline antipsychotic (DUP) | 46 | 248.30 | 245.06 | 0 | 726 |
| Duration from 1st contact with mental health services (days) to baseline antipsychotic | 46 | 346.57 | 600.37 | 6 | 2358 |
| Chlorpromazine equivalents (mg/day) | 46 | 176.89 | 121.29 | 10 | 800 |
| Number of hospital admissions | 46 | 0.89 | 0.64 | 0 | 3 |
| Years of education | 42 | 13.62 | 2.82 | 5 | 20 |
| CGI-SCH baseline score | 56 | Minimally ill=1<br>Mildly ill=4<br>Moderately ill=12<br>Markedly ill=15<br>Severely ill=12<br>Among the most severely ill=1 | – | – | – |
| Antipsychotic medication | 51 | Amisulpride=1<br>Aripiprazole=19<br>Olanzapine=16<br>Paliperidone=1<br>Quetiapine=4<br>Risperidone=5 | – | – | – |

CGI-SCH, Clinician Rating Scale for Schizophrenia; DUP, duration of untreated psychosis.

negative symptoms (table 2). In the whole sample, cognitive performance on the BACS verbal memory significantly improved between baseline and 2-week visits, 2-week and 6-week visits, as well as baseline and 6-week assessments (table 2). Verbal fluency significantly improved between baseline and 2-week visits. Symbol coding, Tower of London and overall (t-score and z-score) performance improved significantly between baseline and 2-week visits and baseline and 6-week visits (table 2).

### Trajectories of symptom change

BIC and AIC values were generated for five classes of trajectory models (table 3). Of these, both indices indicate that the two-trajectory group is best fitted to the data. This model estimated 73.7% of the sampled population to be from one linear trajectory, with the remaining 26.3% in another.

The trajectories identified by the *traj* procedure are shown in figure 2. The trajectories that emerged clearly represented responders versus non-responders. Thirty-nine participants (84.78%) of the sample were classified as antipsychotic treatment responsive and seven as treatment non-responsive (15.22%). For responders, PANSS total score percentage change at 6 weeks was on average 32.89% (SD=27.5) for symptom improvement. For non-responders, this was −21.03% (SD=16.1) indicating a

decline in symptom improvement. Shape estimates and SEs of antipsychotic response are shown in table 4. Treatment responders significantly improved over the 6-week period. Descriptive statistics of clinical and demographic variables between both trajectory groups (non-responder; responder) are presented in the online supplemental material table 4. In comparison with those responsive to antipsychotic medication, non-responders were on average older, had longer duration of time from first contact with mental health services to baseline antipsychotic medication, had marginally more hospital admissions, attained more years of education and were treated at higher chlorpromazine equivalents (online supplemental material table 4).

### Cognitive performance

There was a significant improvement in BACS verbal memory and symbol coding performance between baseline and 6-week assessments across the whole sample, with a significant improvement in Tower of London and BACS z and t composite scores between baseline and 2-week visits (table 2). At baseline assessment, there was no difference in the BACS subscale or composite scores between antipsychotic responders and non-responders identified in the trajectory analysis (tables 5 and 6). Growth curve models observed no significant change

**Table 2** Mean symptom severity as rated by the PANSS and BACS performance for the whole sample for each study visit

| Variable | Baseline | | | 2-week follow-up | | | 6-week follow-up | | | Baseline vs 2 weeks Wilcoxon signed-rank | 2 weeks vs 6 weeks Wilcoxon signed-rank | Baseline vs 6 weeks Wilcoxon signed-rank |
|---|---|---|---|---|---|---|---|---|---|---|---|---|
| | N | M | SD | N | M | SD | N | M | SD | | | |
| PANSS positive | 45 | 11.93 | 4.77 | 36 | 9.17 | 5.36 | 38 | 7.26 | 5.13 | Z=2.76, p=0.006* | Z=2.67, p=0.008* | Z=4.50, p<0.001* |
| PANSS negative | 45 | 10.36 | 6.87 | 36 | 10.31 | 7.15 | 38 | 9.58 | 7.78 | Z=0.78, p=0.435 | Z=0.62, p=0.535 | Z=1.17, p=0.242 |
| PANSS general | 45 | 19.40 | 8.57 | 36 | 16.64 | 10.60 | 38 | 15.74 | 10.11 | Z=3.21, p=0.001* | Z=2.48, p=0.013* | Z=0.64, p=0.524 |
| PANSS total | 45 | 41.69 | 16.11 | 36 | 36.11 | 20.03 | 38 | 32.58 | 20.04 | Z=3.10, p=0.002* | Z=1.46, p=0.144 | Z=3.35, p<0.001* |
| BACS verbal memory | 41 | 37.83 | 14.02 | 32 | 41.16 | 13.50 | 36 | 44.56 | 15.02 | Z=-3.14, p=0.002* | Z=-3.15, p=0.002* | Z=-3.88, p<0.001* |
| BACS digit sequencing | 38 | 18.03 | 4.06 | 32 | 17.84 | 4.33 | 34 | 17.85 | 4.69 | Z=-0.78, p=0.433 | Z=-0.40, p=0.688 | Z=0.40, p=0.687 |
| BACS verbal fluency | 42 | 28.60 | 7.85 | 31 | 31.87 | 9.47 | 35 | 30.20 | 8.38 | Z=-1.96, p=0.050* | Z=0.83, p=0.405 | Z=-1.62, p=0.105 |
| BACS token motor | 39 | 65.36 | 10.83 | 32 | 65.56 | 17.81 | 34 | 61.38 | 20.94 | Z=-1.30, p=0.193 | Z=-0.55, p=0.583 | Z=-0.24, p=0.812 |
| BACS symbol coding | 39 | 40.15 | 13.13 | 32 | 45.69 | 12.56 | 32 | 47.25 | 11.96 | Z=-2.25, p=0.025* | Z=-1.07, p=0.284 | Z=-3.29, p=0.001* |
| BACS ToL | 37 | 14.84 | 4.48 | 29 | 17.38 | 3.29 | 32 | 16.28 | 4.13 | Z=-3.24, p=0.001* | Z=1.45, p=0.148 | Z=-2.42, p=0.016* |
| BACS t-score | 33 | 26.67 | 11.98 | 28 | 34.14 | 11.68 | 30 | 30.87 | 14.95 | Z=-3.79, p<0.001* | Z=-0.29, p=0.769 | Z=-3.66, p<0.001* |
| BACS z-score | 33 | -2.34 | 1.19 | 28 | -1.59 | 1.17 | 30 | -1.91 | 1.50 | Z=-3.85, p<0.001* | Z=-0.23, p=0.820 | Z=-3.67, p<0.001* |

*Significant at p=0.05 level.
BACS, Brief Assessment of Cognition in Schizophrenia; PANSS, Positive and Negative Symptom Scale; ToL, Tower of London.

**Table 3** Selecting a trajectory model using BIC and AIC estimates

| No of classes | 1 | 2 | 3 | 4 | 5 |
|---|---|---|---|---|---|
| BIC | −522.21 | −512.13 | −517.87 | −520.14 | −525.88 |
| AIC | −519.46 | −506.64 | −509.64 | −509.17 | −512.17 |
| % in each class | 100 | 73.7; 26.3 | 73.7; 26.3; 0 | 60.7; 23.7; 15.6; 0 | 60.7; 23.7; 15.6; 0; 0 |

AIC, Akaike's information criterion; BIC, Bayesian information criterion.

in cognitive performance over follow-up visits between trajectory groups (online supplemental material figure 1, table 2). A similar pattern in results was observed when >20% PANSS reduction criteria were applied (online supplemental material figure 2 and table 3,5).

### Multivariable linear regression

Univariable and multivariable logistic regression models adjusting for age and gender and DUP found no significant associations between BACS performance at baseline and response trajectory over 6 weeks (table 6), with no association of any demographic or clinical variables in multivariable models. This was also observed when using the >20% reduction in PANSS total criteria (online supplemental material table 4).

### DISCUSSION

This prospective study investigated the relationship between baseline cognitive performance and subsequent antipsychotic response over a 6-week treatment period. Across the whole sample, participants showed an overall reduction in symptom severity as well as an improvement in cognitive performance on the majority of BACS tasks. Trajectory analyses estimated two trajectory groups

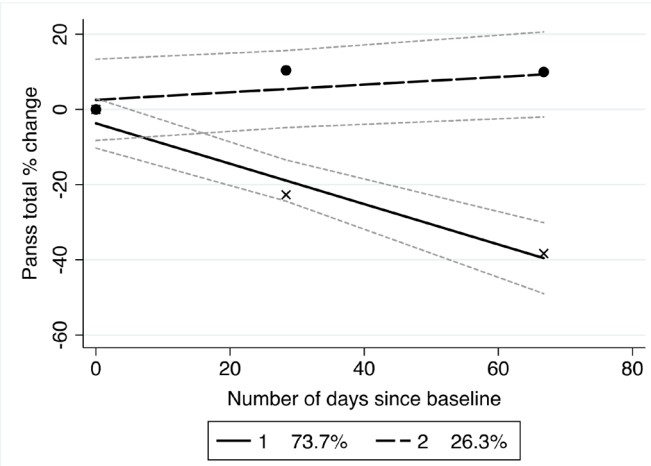

**Figure 2** Trajectory model of total PANSS score percentage change from baseline modelled over days since baseline assessment. The dotted linear trajectory reflects treatment non-responders and the complete line treatment responders. The grey dotted lines around each trajectory reflect the CIs for each trajectory group. Percentages reflect the estimated amount of the sampled population included in each trajectory. PANSS, Positive and Negative Syndrome Scale.

(73.7%, 26.3%) based on PANSS total % change from baseline; this was reflected as 84.78% of the sample being grouped as treatment responsive, and the remaining 15.22% as treatment non-responsive. Contrary to our hypothesis, baseline cognitive performance did not significantly differ between those identified as treatment responders and non-responders following 6 weeks of antipsychotic treatment. This finding remained the same when treatment response was defined as at least a 20% reduction in PANSS total scores, suggesting that there is no association between cognitive performance and antipsychotic response in first-episode schizophrenia.

Across the 2-week and 6-week follow-up visits, an improvement in cognitive performance was observed for the whole sample on BACS measures of verbal memory, verbal fluency, symbol coding and Tower of London tasks, as well as the BACS composite scores. Most of these changes occurred between baseline and 2-week assessment (table 2), with small decreases in performance on measures of verbal fluency, token motor and Tower of London tasks between 2-week and 6-week assessments, as well as for composite z-score and t-score. In contrast, there was a decline in performance in the token motor task across the follow-up period, and minimal changes in performance in the digit sequencing task.

The observed improvement in cognitive performance may reflect a beneficial outcome of antipsychotic treatment. First-generation antipsychotics, introduced in the 1950s, target the positive symptoms observed in schizophrenia by acting as an antagonist at dopamine D2 receptors. Treatment with this group of antipsychotic drugs has been associated with motor and cognitive deficits in patients.[36 37] In contrast, second-generation antipsychotics are reported to have fewer extrapyramidal side effects,[38] with these drugs also acting as an antagonist at the serotonin 5HT2A receptor, in addition to D2 dopamine

**Table 4** Parameter estimates and SEs for both trajectories of antipsychotic response

| | Trajectories | |
|---|---|---|
| Parameters | Non-responder (N=7) | Responder (N=39) |
| Intercept | 2.54 | −3.71 |
| Linear change | 0.10 | −0.54 |
| SE | 0.06 | 0.09 |
| T-statistic | 1.61, p=0.111 | −6.06, p<0.001 |

**Table 5** Baseline cognitive performance for both trajectory groups

| BACS measure | Non-responder | | | Responder | | |
|---|---|---|---|---|---|---|
| | N | Mean | SD | N | Mean | SD |
| Verbal memory | 7 | 37.29 | 9.48 | 34 | 37.94 | 14.89 |
| Digit sequencing | 6 | 20.17 | 5.38 | 32 | 17.63 | 3.74 |
| Verbal fluency | 7 | 30.29 | 7.30 | 35 | 28.26 | 8.01 |
| Token motor | 7 | 66.86 | 8.93 | 32 | 65.03 | 11.30 |
| Symbol coding | 6 | 47.50 | 6.35 | 33 | 38.82 | 13.66 |
| Tower of London | 7 | 14.71 | 3.77 | 30 | 14.87 | 4.69 |
| t-score composite | 6 | 28.83 | 14.36 | 27 | 26.19 | 11.65 |
| z-score composite | 6 | –2.12 | 1.41 | 27 | –2.39 | 1.16 |

BACS, Brief Assessment of Cognition in Schizophrenia.

receptors. Research suggests that in comparison with first-generation, second-generation antipsychotics can provide some improvement in cognitive performance (eg, clozapine[39]). Guilera et al[40] found in their meta-analysis of 18 randomised controlled trials that second-generation drugs provided a slight improvement in performance for global cognition, as well as slight but significant improvements in measures of procedural learning, language and verbal comprehension, verbal learning and memory, and visual learning and memory.

As the whole study sample was treated with second-generation antipsychotic drugs at baseline assessment (amisulpride=1, aripiprazole=19, olanzapine=16, paliperidone=1, quetiapine=4 and risperidone=5), it is possible that the improvement in cognitive performance observed in our sample may be a result of second antipsychotic treatment effects, although first-generation antipsychotic use could not be compared. However, it has also been argued that improvements in cognitive performance over longitudinal designs may instead reflect practice effects (eg, familiarity and procedural learning[41]), meaning that improvement in cognitive performance in our sample could also be attributable to practice effects between

study visits. Lees et al[42] estimated the magnitude of these effects using both the MATRICS Consensus Cognitive Battery (MCCB)[43] and the Cog State Schizophrenia Battery,[44] finding strong test–retest correlations between repeated baseline visits across cognitive batteries, with potential learning effects in socioemotional cognition. However, the authors also observed that participants may have failed to complete the initial baseline assessment due to difficulty in understanding the task, with the suggestion that future investigations using either battery would benefit from adopting initial practice sessions to reduce practice effects. Therefore, an initial practice session with the BACS may have reduced the size of improvement observed in cognitive performance from baseline performance. Another way to determine the extent of practice effects in our sample would be to have a control group who is already stable on antipsychotic medication to see if similar outcomes are observed between groups.

Despite all of the sample being treated with second-generation antipsychotics, it is also possible that some anticholinergic effects, which differ between second-generation antipsychotic drugs,[45] may have affected cognitive performance. Long-term exposure to antipsychotic medications of high anticholinergic activity has been previously reported to impact cognitive performance in patient samples.[46–48] Using low and high anticholinergic activity criteria from a recent review comparing medication effects (from Stroup and Gray[49]; refer to table 1, p342), our sample had 44% (N=20) treated with a high anticholinergic antipsychotic, meaning that the absence of significant differences between groups may have been a result of heterogeneity in medication effects. Therefore, future investigations should consider the role of antipsychotic treatment effects on cognitive outcomes within schizophrenia.

Trajectory analyses identified two clearly defined trajectories of treatment response, both of which are consistent across both time points: one trajectory showing good response, and one of little to no response (figure 2). CIs (figure 2) show some overlap between trajectories

**Table 6** Results from univariable and multivariable logistic regression models for response status and baseline BACS performance

| BACS task | Unadjusted | | | | | Adjusted for age, gender and DUP | | | | |
|---|---|---|---|---|---|---|---|---|---|---|
| | β | SE | 95% CI | OR | P value | β | SE | 95% CI | OR | P value |
| Verbal memory | <0.01 | 0.03 | –0.06 to 0.06 | 1.00 | 0.909 | <–0.01 | 0.03 | –0.07 to 0.06 | 1.00 | 0.918 |
| Digit sequencing | –0.17 | 0.12 | –0.41 to 0.07 | 0.84 | 0.168 | –0.18 | 0.13 | –0.44 to 0.07 | 0.83 | 0.151 |
| Verbal fluency | –0.03 | 0.05 | –0.14 to 0.07 | 0.97 | 0.530 | –0.05 | 0.06 | –0.17 to 0.07 | 0.95 | 0.417 |
| Token motor | –0.02 | 0.04 | –0.09 to 0.06 | 0.98 | 0.683 | –0.0.02 | 0.05 | –0.11 to 0.08 | 0.99 | 0.737 |
| Symbol coding | –0.06 | 0.04 | –0.14 to 0.02 | 0.94 | 0.145 | –0.07 | 0.05 | –0.16 to 0.02 | 0.93 | 0.114 |
| Tower of London | 0.08 | 0.09 | –0.18 to 0.19 | 1.01 | 0.935 | –0.01 | 0.10 | –0.20 to 0.19 | 0.99 | 0.947 |
| t-score composite | –0.02 | 0.04 | –0.10 to 0.06 | 0.98 | 0.620 | –0.02 | 0.04 | –0.10 to 0.06 | 0.98 | 0.594 |
| z-score composite | –0.21 | 0.40 | –0.99 to 0.58 | 0.81 | 0.603 | –0.23 | 0.41 | –1.02 to 0.57 | 0.80 | 0.573 |

BACS, Brief Assessment of Cognition in Schizophrenia; DUP, duration of untreated psychosis.

in the first ~20 days since baseline assessment, with these becoming independent following this period, meaning that separation between trajectory groups was apparent at around 3 weeks. This supports the findings from Samara *et al*[6] who found poor/minimal response to antipsychotic treatment at 2 weeks to be predictive of future treatment non-response. In previous investigations using first episode samples, four or more trajectories have been identified.[50 51] However, both these investigations used longer periods of follow-up as well as raw unadjusted PANSS scores in their analyses: as the minimum raw score of the PANSS is 30, it is recommended rescaling the scores by subtracting 30 from total scores prior to producing percentages and ratios.[52] Therefore, building trajectory models using raw scores may not be appropriate to use as ratio operations (eg, calculating proportions and percentages) require a natural zero point.[52]

Growth curve models, which were used to quantify change in cognitive performance between trajectory groups, observed no significant changes in performance between visits. It is possible that this may be due to undersampled groups, as significant improvements for verbal memory, symbol coding, Tower of London and composite scores were observed in the whole sample. When comparing our findings with a >20% reduction in rescaled PANSS total score criteria,[24 25] there were no changes in the pattern of results to growth curve models or logistic regression outcomes. Using this criterion for treatment response resulted in a more even distribution of the total sample to groups (responder=17; non-responder=21), providing more power to comparative analyses. However, despite this, there was no change in the pattern of results, meaning that this criterion provided no added benefit to this analysis over trajectory-based groupings. The lack of significant difference in baseline cognition between those classified as treatment responders and non-responders after 6 weeks of treatment in our study contrasts previous research conducted, which observed impaired cognitive performance in the poor response trajectory at week 4, with good performance at baseline being predictive of a good response trajectory at week 4.[51] Likewise, longitudinal research using the MCCB[43] with patients with first-episode schizophrenia assessed at baseline and at a 12-week follow-up identified tasks of executive function and planning and reasoning ability as potential indices of antipsychotic response,[53] with similar findings observed when cognitive performance is correlated with symptom severity measures.[54]

## Limitations

Previous investigations included sample sizes several magnitudes higher than in our study (Levine and Rabinowitz,[51] N=491; Trampush *et al*,[53] N=175) and it is likely that our sample size limited our ability to observe a significant relationship between cognitive performance and antipsychotic response. Using our sample's mean values for the BACS t composite score, a power calculation found that a total sample size of 31 304 samples would be

required to detect a significant difference between trajectory groups at 90% power. When using the >20% PANSS reduction criteria, this was N=6118, suggesting that both analyses were underpowered due to undersampling.

Another considerable limitation of the conclusions from this investigation is the expectation of detecting meaningful change in both clinical response to medication and cognition in such short duration of follow-up. Previous longitudinal investigations into cognitive change have noted that even a period of 1–3 years may not be substantial to detect changes in cognitive performance,[55] questioning the additional analyses in this study comparing performance between baseline and 2-week and 6-week study visits. Likewise, Emsley *et al*'s[56] investigation with 522 participants with first-episode schizophrenia found 11.2% of their sample to not achieve clinical response (determined by a 20% improvement in PANSS total scores) until after 8 weeks, with the authors concluding that antipsychotic response is greatly varied and that longer investigations are needed to capture the large variability in clinical response.[56] Therefore, it is also possible that there are participants within the sample who may have later responded to medication if the follow-up was at longer duration, which may also partially support the lack of significant differences between groups in this study. Likewise, adopting secondary criteria for treatment response and non-response based on criteria from the Treatment-Resistant Schizophrenia: Treatment Response and Resistance in Psychosis Working Group[57] would also help in seeing whether the groupings identified by trajectory analyses correspond to standardised guidelines, aiding in comparison between investigations.

Due to the issues with small sample sizes, it was not possible to adjust for additional variables, which may be associated with cognitive performance. Negative symptoms have routinely been associated with cognitive performance,[58 59] including performance on the BACS.[60] Medication effects, such as higher antipsychotic doses[61 62] and high anticholinergic antipsychotics,[46–48] have also been associated with deficits in cognitive performance. Future research should measure and adjust for these variables in order to determine the true association between cognition and treatment response without potential confounders.

It is also possible that premorbid histories of the sample may have resulted in a less consistent picture of cognitive performance between groups. For example, prior cannabis use, particularly during adolescence, has been found to improve cognitive performance on the BACS in comparison with those who have not ever used cannabis.[63] In this investigation comparing performance on the BACS between patients with a schizophrenia diagnosis with and without adolescent cannabis use (ACU), those with ACU reported significantly higher composite scores, as well significant improvement in working memory and verbal memory tasks.[63] In our sample, 68% (N=30) had previous experience of using cannabis, with the majority of this use occurring between ages 12 and

19 years (N=23). Therefore, it is possible that premorbid histories may have also blurred the cognitive differences between groups.

## CONCLUSIONS

In this prospective cohort study, patients with a first-episode diagnosis were assessed three times over a period of 6 weeks. Trajectory analyses using percentage change in PANSS total symptom scores identified two groups reflecting a good and poor response to antipsychotic medication. Baseline cognitive performance of these two groups did not predict response status at 6 weeks. This lack of discrimination between groups is potentially attributable to underpowered analyses as a result of small sample sizes but may also evidence that an association between cognition and treatment response is not observable in the first episode of schizophrenia. Overall, this suggests that brief cognitive batteries for schizophrenia may not be a useful predictor of antipsychotic response in the first 2 years of illness onset.

**Author affiliations**
[1]Department of Psychosis Studies, King's College London Institute of Psychiatry, Psychology and Neuroscience, London, UK
[2]NIHR Biomedical Research Centre, South London and Maudsley NHS Foundation Trust, London, UK
[3]National Psychosis Service, South London and Maudsley Mental Health NHS Trust, London, UK
[4]Division of Psychology and Mental Health, University of Manchester, Manchester, UK
[5]Research and Innovation, Greater Manchester Mental Health NHS Foundation Trust, Manchester, UK
[6]Pharmacy Department, South London and Maudsley NHS Foundation Trust, London, UK
[7]Department of Epidemiology & Applied Clinical Research, University College London, London, UK
[8]Division of Psychological Medicine and Clinical Neurosciences, Cardiff University, Cardiff, UK
[9]Sackler Faculty of Medicine, Shalvata Mental Health Center, Hod Hasharon, Israel

**Contributors** JM, AE, BD, RD, ODH, LK, CC, AS, RO, SLe, JL, SLa, SK and EM contributed to the design and implementation of the study. EO, EM, RP, NR, KG, CC, SS and KVS aided in data collection. EM completed analyses and wrote the manuscript with the assistance of JM, KG, NR, AE, EK, RD, AS, CC and BD, and provided comments on the manuscript. JM acted as guarantor.

**Funding** STRATA is funded by a grant from the Medical Research Council (MRC) to JM (grant reference MR/L011794). EM's PhD is funded by the MRC-doctoral training partnership studentship in Biomedical Sciences at King's College London. JM, EK, AE and ODH are part funded by the National Institute for Health Research (NIHR) Biomedical Research Centre at South London and Maudsley NHS Foundation Trust and King's College London.

**Disclaimer** The views expressed are those of the authors and not necessarily those of the NHS, the MRC, the NIHR or the Department of Health.

**Competing interests** None declared.

**Patient and public involvement** Patients and/or the public were involved in the design, or conduct, or reporting, or dissemination plans of this research. Refer to the Methods section for further details.

**Patient consent for publication** Obtained.

**Ethics approval** This study involves human participants and was approved by the Health and Social Care Research Ethics Committee A (REC: 17/NI/0209). All participants provided informed consent prior to participation.

**Provenance and peer review** Not commissioned; externally peer reviewed.

**Data availability statement** Data are available upon reasonable request. At the time of submission, the data governance frameworks are being put in place to make a fully anonymised version of the data available to the wider research community via the TranSMART data sharing platform (https://transmartfoundation. org/). To apply for access to the data, please contact JM at james.maccabe@kcl. ac.uk.

**ORCID iDs**
Edward Millgate http://orcid.org/0000-0001-5424-8261
Richard Drake http://orcid.org/0000-0003-0220-4835
Rebecca Pollard http://orcid.org/0000-0002-6516-6891

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
