## [Reviewer comments · BMJ Open]

ARTICLE DETAILS

TITLE (PROVISIONAL)	Cognitive function and treatment response trajectories in first episode schizophrenia: evidence from a prospective cohort study.
AUTHORS	Millgate, Edward; Griffiths, Kira; Egerton, Alice; Kravariti, Eugenia; Casetta, Cecilia; Deakin, Bill; Drake, Richard; Howes, Oliver; Kassoumeri, Laura; Khan, Sobia; Lankshear, Steve; Lees, Jane; Lewis, Shon; Mikulskaya, Elena; Oloyede, Ebenezer; Owens, Rebecca; Pollard, Rebecca; Rich, Nathalie; Smart, Sophie; Segev, Aviv; Verena Sendt, Kyra; MacCabe, James

VERSION 1 – REVIEW

REVIEWER	Chong, Catherine Hospital Authority Head Office
REVIEW RETURNED	26-Apr-2022

GENERAL COMMENTS	The study question per se is interesting. The statistical methods used were robust. However, with the very small sample size, it is very difficult to differentiate the negative finding to be truly negative or a result of an underpowered study. As such this article may not add too much value to the existing literature.
---

REVIEWER	Beaudoin, Mélissa University of Montreal, Psychiatry and addictology
REVIEW RETURNED	23-May-2022

GENERAL COMMENTS	This paper reports the findings of a small longitudinal study investigating a possible association between cognitive function and subsequent antipsychotic response among people with first-episode psychosis. Forty-six participants were assessed three times over 6 weeks and treatment response trajectories were calculated based on the PANSS, whereas functioning was measured using the BACS. However, no significant association was observed. The authors suggest that it might be due to a lack of statistical power, or to the fact that brief cognitive batteries might not be useful predictors of treatment response in that population. Despite the results being inconclusive, the hypothesis remains interesting, and publishing such negative results is important. Nevertheless, I have some major and less major concerns; please find my comments and questions below (per section). Abstract & Article summary 1. It could easily be argued that N=46 is not a “relatively large” sample; compared to similar studies, the sample is small.2. “Trajectory analyses used identified two clear patterns”: the English language should be carefully revised throughout the
--

	article, as such mistakes are common and can hinder the reader's understanding. 3. It would have been relevant to indicate whether there is a tendency towards significance or not, and if so, what is the effect size? Introduction 4. It would have been relevant to discuss a bit more the importance of this study, and how it was expected to contribute to the existing literature. The current introduction is very short compared to the other sections and it could definitely be more detailed. Methods 5. Participants: It would be very relevant to include a CONSORT flowchart so that the reader could see how many participants were assessed at each timepoint. It would also be interesting to see how many participants were excluded for each reason separately. 6. Is the baseline visit happening +/- 7 days following antipsychotic initiation? This is currently unclear and should be specified. 7. " with the maximum cut-off for 6-week follow up being 78 days after baseline assessment.": I am unsure about what the authors mean by "cut-off", but if they are saying that the maximum number of days elapsed between the baseline and the 6-week visit was 78 days, they should explain how and why that happened. Six weeks represents 42 days, and even when accounting for the +/- 7 days, participants should not be assessed more than 49 days after baseline. The authors should minimally mention how many participants were assessed past that time, and ideally also provide, a mean, a median and/or a standard deviation for the number of days elapsed between each visit. 8. Definitions for treatment response status: Although this is later detailed in the Discussion section, the authors should specify what they mean by "Rescaled PANSS scores" directly in the Methods section. 9. Neuropsychological assessment: I am wondering why the authors did not include a practice session prior to the baseline. This should be justified as it greatly undermines the credibility of the results. 10. Data analysis: The authors decided to use independent t-tests to compare cognitive performance and symptom severity between visits. I have two major concerns regarding that decision:  Did the authors make sure that the data was normally distributed? I suggest testing this using a Shapiro-Wilk or Kolmogorov-Smirnov test. If this is not the case, a Mann-Whitney test would be more appropriate. As these measurements were repeated over time for the same participants, a paired sample t-test should be performed, as these mean scores are not independent. Results 11. Adding a symbol (e.g., *) when a result is statistically significant would make the reading of the tables much easier. 12. "-32.89% symptom improvement": the negative makes the sentence very confusing. Is this a 32% improvement or a -32% variation? 13. "For non-responders this was 21.03% indicating a minimal and, in some cases, worsening in symptom severity.": is this the average symptom improvement? If so, I am confused as to why the authors say its "minimal" since they define a clinically
--	---

	significant improvement as being above 20%. Moreover, it would be relevant to provide the standard deviation and range for these improvement values. Discussion 14. The discussion should be revised once the appropriate statistical tests will be made, as independent t-tests are not appropriate for that design. 15. Regarding the lack of practice session: one way to account for this effect would be to add a control group who's been stable on a medication for a while. 16. Regarding rescaling the PANSS: I do not understand how not rescaling the PANSS could change your results since, unless I am mistaken, none of your statistics include ratios? 17. In the Introduction, the authors mention that what is observed in the current literature on the association between neurocognition and treatment response could be explained by illness chronicity and long-term antipsychotic treatment. That hypothesis is consistent with the fact that this association does not seem to be present in their sample of first-episode psychosis. This would be interesting to discuss this matter in the Discussion section. 18. Limitations: The authors argued that the lack of statistically significant results might be due to the small sample size. However, even if a sample size of several thousands of participants would yield significant results, it is very questionable whether these results would have any clinical significance at all. With such a small effect size, that seems unlikely. Conclusions 19. I disagree with the authors stating that the lack of significance of the results is "likely" due to the analyses being underpowered; by looking at the results, it simply seems that the association is very weak or completely inexistent. Even if having an enormous sample size might yield significant results, this does not mean that these would be clinically meaningful. Supplementary material 20. Table S.1: in the non-responder column, the mean number of hospitalizations is 100; this seems to be an error, as the authors probably meant to write 1.00. Please also verify the SD of that line as its value of 1.00 might have been misplaced.
--	--

VERSION 1 – AUTHOR RESPONSE

Reviewer #1:

General comment:

The study question per se is interesting. The statistical methods used were robust. However, with the very small sample size, it is very difficult to differentiate the negative finding to be truly negative or a result of an underpowered study. As such this article may not add too much value to the existing literature.

AUTHORS' RESPONSE: We thank Reviewer 1 for this review of our submitted manuscript. While we agree that the sample size, which was based off power calculations needed to detect differences in neuroimaging metabolites (i.e. glutamate in the anterior cingulate cortex), we argue that the submitted manuscript is an important addition to the existing literature. This is because the study adds further discussion towards the association, or perhaps lack of association as found in this study, and

treatment response in schizophrenia. The manuscript also uses methodologically sound methods for analysis and recruitment which should be considered as a benchmark for future research to expand on this research in larger samples. Finally, the data included throughout the main manuscript and supplementary is also of beneficial use for future meta-analyses and systematic reviews. Our corrections following the recommendations to Reviewer 2 hopefully highlight the importance of this manuscript as well.

Reviewer #2:

General comment:

This paper reports the findings of a small longitudinal study investigating a possible association between cognitive function and subsequent antipsychotic response among people with first-episode psychosis. Forty-six participants were assessed three times over 6 weeks and treatment response trajectories were calculated based on the PANSS, whereas functioning was measured using the BACS. However, no significant association was observed. The authors suggest that it might be due to a lack of statistical power, or to the fact that brief cognitive batteries might not be useful predictors of treatment response in that population.

Despite the results being inconclusive, the hypothesis remains interesting, and publishing such negative results is important. Nevertheless, I have some major and less major concerns; please find my comments and questions below (per section).

AUTHORS' RESPONSE: We thank Reviewer 2 for their kind comments and summation of the research presented in our manuscript. We are pleased that the hypothesis of this research is of interest, and we look forward to addressing your comments to each section below.

Specific comments:

1.

Reviewer #2:

Abstract & Article summary

- a. It could easily be argued that N=46 is not a "relatively large" sample; compared to similar studies, the sample is small.
- b. "Trajectory analyses used identified two clear patterns": the English language should be carefully revised throughout the article, as such mistakes are common and can hinder the reader's understanding.
- c. It would have been relevant to indicate whether there is a tendency towards significance or not, and if so, what is the effect size?

AUTHORS' RESPONSE: We thank Reviewer 2 for these comments on the abstract and article summary sections.

For the Article summary section, we agree that 46 participants do not equate to a "relatively large" sample for this investigation, so this was removed from the summary section. Likewise, the grammar in the second bullet point of the Article summary ("Trajectory analyses used...") was corrected. Further grammatical errors were also checked and changed throughout the main manuscript to prevent hindering the reader's understanding.

For the Abstract, we included additional information and results to the *Results* section so that we are explicit about the lack of trend for significance in our results: "Unadjusted and adjusted logistic regressions observed no significant relationship between baseline BACS on subscale and total performance (BACS t-score: OR = 0.98, $p = .620$, Cohen's $d = .218$) and antipsychotic response at 6-weeks."

2.

Reviewer #2:

Introduction

d. It would have been relevant to discuss a bit more the importance of this study, and how it was expected to contribute to the existing literature. The current introduction is very short compared to the other sections and it could definitely be more detailed.

AUTHORS' RESPONSE: We are thankful to Reviewer 2 for showing interest in the study subject area and their recommendation to expand on the importance and contribution of this study to the wider academic and scientific community. The following paragraphs were included to the introduction section on pages 5 and 6 respectively:

"Individuals who do not respond to antipsychotic medication are reported to have higher rates of smoking (56%), substance and alcohol abuse (51%) and suicidal ideation (44%), with annual treatment costs being 3 to 11 times larger than those who respond to antipsychotic medication⁹. In 2007, it was estimated that schizophrenia accounted for 30% of the total expenditure for adult mental health and social care services¹⁰, with additional economic and societal costs due to unemployment or absence from work. These total service costs, which were estimated at £2.2 billion in 2007, have the potential to reach £3.7 billion by 2026¹¹. However, it has been suggested that early intervention programmes could aid in reducing these costs substantially if adequately introduced in first episode psychosis¹², as earlier onset schizophrenia is associated with greater expected costs¹¹."

"Based on the current existing literature it is plausible to argue that there may be quantifiable cognitive differences between individuals who respond to antipsychotic medication and those who do not in the early stages of the illness; seeing as deficits in cognition are observable prior to illness onset^{14,15} and poor early non-response to medication being predictive of future non-response⁵. Therefore if differences are observed between groups of differing response to medication (i.e. responders and non-responders), early in their illness and treatment, this will broaden our understanding of the relationship between cognition, schizophrenia, and antipsychotic response, as well as aid clinical utility by using brief cognitive measures as a screening for potential non-response in the first episode of schizophrenia. The American Psychological Association's Working Group on Screening and Assessment have provided guidelines for determining the appropriateness of a neuropsychological measure for cognitive screening within a clinical setting¹⁹. The guidelines are as follows: i. provide identification for those at high risk for impairment, ii. sensitive enough to identify those who need further review, iii. brief and narrow in scope, iv. can be administered at routine visits, v. can be administered by support staff or clinicians electronically and vi. can be used to monitor progress and

outcomes²⁰. In high-income countries, the use of brief assessment batteries such as the BACS have been found to meet these criteria put forward by the APA working group²¹.”

3.

Reviewer #2:

Methods

e. Participants: It would be very relevant to include a CONSORT flowchart so that the reader could see how many participants were assessed at each timepoint. It would also be interesting to see how many participants were excluded for each reason separately.

f. Is the baseline visit happening +/- 7 days following antipsychotic initiation? This is currently unclear and should be specified.

g. “ with the maximum cut-off for 6-week follow up being 78 days after baseline assessment.”: I am unsure about what the authors mean by “cut-off”, but if they are saying that the maximum number of days elapsed between the baseline and the 6-week visit was 78 days, they should explain how and why that happened. Six weeks represents 42 days, and even when accounting for the +/- 7 days, participants should not be assessed more than 49 days after baseline. The authors should minimally mention how many participants were assessed past that time, and ideally also provide, a mean, a median and/or a standard deviation for the number of days elapsed between each visit.

h. Definitions for treatment response status: Although this is later detailed in the Discussion section, the authors should specify what they mean by “Rescaled PANSS scores” directly in the Methods section.

i. Neuropsychological assessment: I am wondering why the authors did not include a practice session prior to the baseline. This should be justified as it greatly undermines the credibility of the results.

j. Data analysis: The authors decided to use independent t-tests to compare cognitive performance and symptom severity between visits. I have two major concerns regarding that decision:

aa. Did the authors make sure that the data was normally distributed? I suggest testing this using a Shapiro-Wilk or Kolmogorov-Smirnov test. If this is not the case, a Mann-Whitney test would be more appropriate.

bb. As these measurements were repeated over time for the same participants, a paired sample t-test should be performed, as these mean scores are not independent.

AUTHORS' RESPONSE: We are grateful for the recommendations made by Reviewer 2 to further improve our methods section of this manuscript. For this section we will respond to these comments on a point-by-point basis for clarity:

- e. A CONSORT flowchart was added on pg.9 of the methods section to illustrate the numbers of participants included at each time point and for which reason. Please find the following flowchart below:

- Regarding comment f. baseline occurred within the first 2-weeks of antipsychotic medication initiation. To make this clearer, the following was added to the *Participants* section of the methods:

“...after a period of being either antipsychotic naïve or antipsychotic-free for at least 14 days. Participants were assessed at baseline within the first 2 weeks of antipsychotic medication initiation.”

“Participants were assessed within the first 14 days of starting antipsychotic medication at baseline, 2-weeks from baseline assessment (± 7 days of date) and 6-weeks...”

- Regarding comment g. the follow-up at 2-week and 6-week visits from baseline allowed a variation of ± 7 days for each visit. However we agree with Reviewer 2 that the maximum number of days has been overestimated as the baseline visit was also included in this estimation. The following was updated in the methods section to make this clearer:

“Participants were assessed within the first 14 days of starting antipsychotic medication at baseline, 2-weeks from baseline assessment (± 7 days of date) and 6-weeks from baseline assessment (± 7 days of date), with the maximum cut-off for 6-week follow up being 56 days after baseline assessment (i.e. if an individual was assessed at the maximum follow-up periods at 2-week and 6-week visits; 8-weeks total).”

In addition to this, we also included the average follow-up in days for each visit, as well as the total trial duration, to the beginning of the Results section:

“Between baseline and 2-week assessment the average follow-up was 18.19 days (SD = 6.6) and between 2-week and 6-week this was 26.69 days (SD = 9.6). Between baseline and 6-week visit, the study trial lasted 43.86 days (SD = 7.2).”

- Regarding comment h. the following was added to the *Definitions of treatment response status section* of the methods to explain what a rescaled PANSS score means:

“...were assessed for eligibility. Rescaled PANSS scores¹⁷ were calculated by subtracting 30 from total scores prior to producing estimates for percentage change a...”

- In response to comment i. from Reviewer 2, it was unfortunate that we were unable to implement a training set of the BACS prior to participation. This may indeed explain why significant improvement in BACS performance was observed between baseline and follow-up visits, however it is also possible that this is a reflection of the beneficial effects of second-generation antipsychotics on cognitive performance. We have commented on this issue in the discussion section:

“However it has also been argued that improvements in cognitive performance over longitudinal designs may instead reflect practice effects (e.g. familiarity and procedural learning⁴¹), meaning that improvement in cognitive performance in our sample could also be attributable to practice effects between study visits. Lees et al⁴² estimated the magnitude of these effects using both the MATRICS

Consensus Cognitive Battery (MCCB)⁴³ and the Cog State Schizophrenia Battery⁴⁴, finding strong test-retest correlations between repeated baseline visits across cognitive batteries, with potential learning effects in social-emotional cognition. However, the authors also observed that participants may have failed to complete the initial baseline assessment due to difficulty in understanding the task, with the suggestion that future investigations using either battery would benefit from adopting initial practice sessions to reduce practice effects. Therefore an initial practice session with the BACS may have reduced the size of improvement observed in cognitive performance from baseline performance.”

- Regarding point j. we agree with the concerns of Reviewer 2 of the use of independent t-tests to test for cognitive and symptom severity differences between study visits. The majority of variables were normally distributed as per Shapiro-Wilk estimates however this was not consistent throughout. In light of this we have updated the results from Table 2 using Wilcoxon signed rank tests, and included the following to the *Data analysis* section of the methods:

“All analyses were conducted in STATA 15/SE²⁷. Wilcoxon signed rank tests were used to compare cognitive performance and symptom severity in the whole sample between visits (i.e. baseline assessment to 2-week, 2-week to 6-week, and baseline to 6-week) as not all symptom severity and cognitive variables were normally distributed.”

The following was also added to the results section to denote the significant findings from these Wilcoxon signed rank tests:

“Between study visits, a significant improvement in PANSS positive symptoms scores was observed in the whole sample between baseline and 2-week visits, 2-week and 6-week visits, as well as baseline and 6-week assessments (Table 2). A significant improvement in PANSS total scores was observed between baseline and 2-week and baseline and 6-week visits. No significant differences in symptom severity were observed between visits for negative symptoms (Table 2). In the whole sample, cognitive performance on the BACS verbal memory significantly improved between baseline and 2-week visits, 2-week and 6-week visits, as well as baseline and 6-week assessments (Table 2). Verbal fluency significantly improved between baseline and 2-week visits. Symbol coding, Tower of London, and overall (t-score and z-score) performance improved significantly between baseline and 2-week visits and baseline and 6-week visits (Table 2).”

4.

Reviewer #2:

Results

k. Adding a symbol (e.g., *) when a result is statistically significant would make the reading of the tables much easier.

l. “-32.89% symptom improvement”: the negative makes the sentence very confusing. Is this a 32% improvement or a -32% variation?

m. “For non-responders this was 21.03% indicating a minimal and, in some cases, worsening in symptom severity.”: is this the average symptom improvement? If so, I am confused as to why the authors say its “minimal” since they define a clinically significant improvement as being above 20%. Moreover, it would be relevant to provide the standard deviation and range for these improvement values.

AUTHORS’ RESPONSE: We thank Reviewer 2 for their suggestions to our results section. To Table 2, the only table reporting significant differences, asterisks were added to the significant findings from the independent t-tests. The reasons for why we had “-32.89% symptom improvement” is because the PANSS is scored as higher scores denoting higher/worse symptoms. Therefore we originally kept the reporting of scores to reflect this, however it does encompass a shroud of confusion. Due to this we have updated these scores to reflect the context of the sentence as well as included standard deviation values for each estimate:

“For responders, PANSS total score percentage change at 6 weeks was on average 32.89% (SD = 27.5) symptom improvement. For non-responders this was -21.03% (SD = 16.1) indicating a minimal and, in some cases, worsening in symptom severity.”

5.

Reviewer #2:

Discussion

n. The discussion should be revised once the appropriate statistical tests will be made, as independent t-tests are not appropriate for that design.

o. Regarding the lack of practice session: one way to account for this effect would be to add a control group who’s been stable on a medication for a while.

p. Regarding rescaling the PANSS: I do not understand how not rescaling the PANSS could change your results since, unless I am mistaken, none of your statistics include ratios?

q. In the Introduction, the authors mention that what is observed in the current literature on the association between neurocognition and treatment response could be explained by illness chronicity and long-term antipsychotic treatment. That hypothesis is consistent with the fact that this association does not seem to be present in their sample of first-episode psychosis. This would be interesting to discuss this matter in the Discussion section.

r. Limitations: The authors argued that the lack of statistically significant results might be due to the small sample size. However, even if a sample size of several thousands of participants would yield

significant results, it is very questionable whether these results would have any clinical significance at all. With such a small effect size, that seems unlikely.

AUTHORS' RESPONSE: We thank Reviewer 2 for these recommendations to improve the discussion section of this manuscript. Following the updates to the analysis (i.e. Wilcoxon signed rank test; Table 2), the results reported in the discussion chapter have been updated. Regarding your recommendation about practice effects, this idea was also added to the discussion section:

“Therefore an initial practice session with the BACS may have reduced the size of improvement observed in cognitive performance from baseline performance. Another way to determine the extent of practice effects in our sample would be to have a control group who is already stable on antipsychotic medication to see if similar outcomes are observed between groups.”

Regarding comment p. ratios (i.e. percentages in this case) were used in our analysis as part of the comparison between criteria for trajectory models and the 20% improvement in PANSS total scores as well as were used for trajectory analyses. Due to this rescaling PANSS scores was necessary in order to carry out this comparison.

In light of comments q and r. we agree that simply arguing the lack of significant findings to be attributable to the small sample size is a bit reductionist, as these findings may also infer that no relationship between antipsychotic response and cognition exists in first episode schizophrenia. Based on this, the following was added to the discussion section:

“20% reduction in PANSS total scores, suggesting that there is no association between cognitive performance and antipsychotic response in first episode schizophrenia.”

As well as further explored more potential limitations in the *limitations* section:

“Another considerable limitation of the conclusions from this investigation is the expectation of detecting meaningful change in both clinical response to medication and cognition in such a short duration of follow-up. Previous longitudinal investigations into cognitive change have noted that even a period of 1 to 3 years may not be substantial to detect changes in cognitive performance⁵⁵, questioning the additional analyses in this study comparing performance between baseline and 2-week and 6-week study visits. Likewise, Emsley et al's⁵⁶ investigation with 522 participants with first episode schizophrenia found 11.2% of their sample to not achieve clinical response (determined by a 20% improvement in PANSS total scores) until after 8 weeks, with the authors concluding that antipsychotic response is greatly varied and that longer investigations are needed to capture the large variability in clinical response⁵⁶. Therefore it is also possible that there are participants within the sample who may have later responded to medication if the follow-up was at a longer duration, which may also partially support the lack of significant differences between groups in this study. Likewise, adopting secondary criteria for treatment response and non-response based off criteria from the TRIPP Working Group⁵⁷ would also help in seeing whether the groupings identified by trajectory analyses correspond to standardised guidelines, aiding in comparison between investigations.

Due to the issues with small sample sizes, it was not possible to adjust for additional variables which may be associated with cognitive performance. Negative symptoms have routinely been associated with cognitive performance^{58,59}, including performance on the BACS⁶⁰. Medication effects such as higher antipsychotic doses^{61,62} and high anticholinergic antipsychotics^{46,47,48}, have also been associated with deficits in cognitive performance. Future research should measure and adjust for these variables in order to determine the true association between cognition and treatment response without potential confounders.”

6.

Reviewer #2:

Conclusions

s. I disagree with the authors stating that the lack of significance of the results is “likely” due to the analyses being underpowered; by looking at the results, it simply seems that the association is very weak or completely in-existent. Even if having an enormous sample size might yield significant results, this does not mean that these would be clinically meaningful.

AUTHORS’ RESPONSE: We agree with the recommendation from Reviewer 2 to revise our conclusions so that they accurately reflect the results and discussion in this manuscript. As such as the following was changed in the conclusions paragraph:

“This lack of discrimination between groups is potentially attributable to underpowered analyses as a result of small sample sizes but may also evidence that an association between cognition and treatment response is not observable in the first episode of schizophrenia. Overall this suggests that brief cognitive batteries for schizophrenia may not be a useful predictor of antipsychotic response in the first two-years of illness onset.”

7.

Reviewer #2:

Supplementary material

t. Table S.1: in the non-responder column, the mean number of hospitalizations is 100; this seems to be an error, as the authors probably meant to write 1.00. Please also verify the SD of that line as its value of 1.00 might have been misplaced.

AUTHORS’ RESPONSE: We thank Reviewer 2 for also taking the time to check our supplementary materials as well. We can confirm that there was an error of missing a full stop (i.e. 1.00 instead of

100), this has been updated in the supplementary table (which is now Table S.4). The SD with a value of 1.00 for the same line is in line with the results from the study.

VERSION 2 – REVIEW

REVIEWER	Beaudoin, Mélissa University of Montreal, Psychiatry and addictology
REVIEW RETURNED	12-Aug-2022

GENERAL COMMENTS	Overall, I am very pleased with the changes the authors made to the manuscript. I have one last very minor comment. Page 15 lines 43-45: The authors improved this sentence, but it remains unclear. A mean variation of -21.03% probably indicates a deterioration of symptoms in most participants, not only "some cases". Moreover, it seems like a word is missing: "indicating a minimal [...response?], and in some cases, worsening symptom severity.". This is very minor, but this hinders the reader's comprehension, and therefore this sentence should be rewritten and clarified. Once this small change will be made, I believe this manuscript will be suitable for publication, and therefore recommend its acceptance.
--